# Eidetic Learning: an Efficient and Provable Solution to Catastrophic Forgetting

## Abstract

Catastrophic forgetting – the phenomenon of a neural network learning a task $t_1$ and losing the ability to perform it after being trained on some other task $t_2$ – is a long-standing problem for neural networks (McCloskey & Cohen, 1989). We present a method, Eidetic Learning, that provably solves catastrophic forgetting. A network trained with Eidetic Learning – here, an EideticNet – requires no rehearsal or replay. We consider successive discrete tasks and show how at inference time an EideticNet automatically routes new instances without auxiliary task information. An EideticNet bears a family resemblance to the sparsely-gated Mixture-of-Experts layer Shazeer et al. (2016) in that network capacity is partitioned across tasks and the network itself performs data-conditional routing. An EideticNet is easy to implement and train, is efficient, and has time and space complexity linear in the number of parameters. The guarantee of our method holds for normalization layers of modern neural networks during both pre-training and fine-tuning. We show with a variety of network architectures and sets of tasks that EideticNets are immune to forgetting. While the practical benefits of EideticNets are substantial, we believe they can be benefit practitioners and theorists alike. We will release the code repository containing the EideticNet PyTorch framework upon publication.[1]

## 1 Introduction

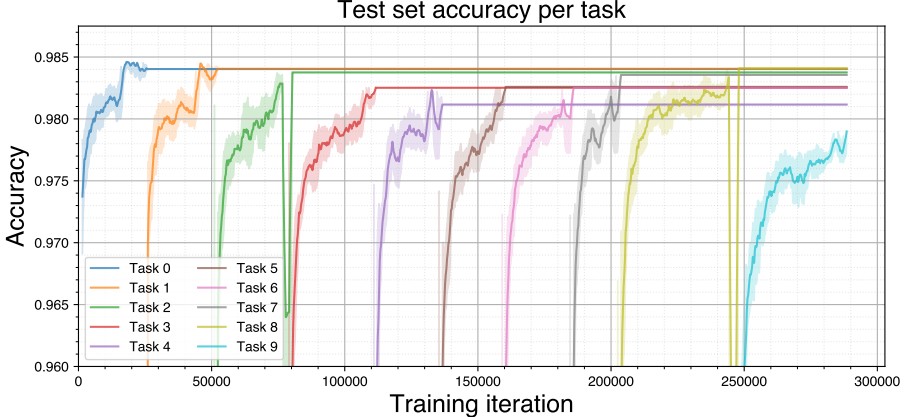

Figure 1: Accuracy of an MLP trained on 10 tasks of Permuted MNIST in a single run of our method. Lines (bands) are a moving average (standard deviation) over a window of 10 steps. See Table 3 for a comparison with other methods.

While artificial neural networks (ANNs) have been demonstrated time and again to be effective devices for learning tasks, they are flawed in task retention. The tendency to lose the ability to

---

[1]Note to reviewers: we are awaiting the completion of the legal review process, so are not yet able to provide a URL for our repository.

perform a task $t_1$ after being trained on a subsequent task $t_{i>1}$ – *catastrophic forgetting* – is a long-standing problem for neural networks (Rumelhart et al., 1986; McClelland et al., 1987; McCloskey & Cohen, 1989).

Many deep neural networks are overparameterized, or have *excess capacity*, for a given task. This is evinced by their ability to minimize training loss even under a random labeling of a training set (Zhang et al., 2017; 2021). In this work, we demonstrate that excess capacity can be exploited to prevent catastrophic forgetting. We present a method, Eidetic Learning, that uses an ANN's excess capacity to guarantee that an ANN trained this way – an EideticNet – does not forget. Eidetic Learning is so named because eidetic memory is perfect recall – typically of visual stimuli, although Eidetic Learning's guarantees hold for all modalities of input data. Figure 1 illustrates the correctness of our method and its implementation.

Eidetic Training exploits iterative pruning (Han et al., 2015): after training a task to convergence, prune neurons until *training set* accuracy drops below a threshold. Then freeze the unpruned neurons, delete the synapses from pruned to unpruned neurons (directionally), and recycle the pruned neurons for subsequent tasks. We employ structured pruning and select neurons to prune using $\ell_1$ or $\ell_2$ weight magnitude pruning, or Taylor pruning (Molchanov et al., 2017).

Figure 2 illustrates the architectural simplicity of EideticNets. A naive partitioning of important neurons for task $t_i$ from unimportant neurons for the task results in disjoint subnetworks and deprives tasks $t_{j>i}$ the opportunity to learn more efficiently by reusing features learned during $t_i$ (Figure 2a). EideticNets enable this more efficient learning by allowing subsequent tasks to benefit from features learned during training of previous tasks (Figure 2b)[2]. There may be some value to defining an EideticNet consisting of both nested and disjoint subnetworks – imagine, for instance, several nested layers followed by one or more disjoint layers – but an evaluation of this possibility is out of the scope of the current work.

Eidetic Learning does not require rehearsal or replay to maintain or improve performance on past or future tasks. However, to eliminate the need for task IDs at inference time, it trains a final task classifier on a *meta task dataset*. The meta task dataset is constructed from all data $(x_{i_t}, y_{i_t}) \, \forall t \in \{1, \ldots, T\}$ from already-trained tasks by seteting the targets $y_{i_t}$ of each specific task $t$ to $t$. At inference time, the task ID is obtained from the task classifier and the appropriate classifier for the instance is invoked. This is done in a single pass of the body of the network. The hidden states of the ANN's penultimate layer are passed first through the task-classification head, then through the selected classifier for the instance. Alternative approaches, such as passing a new instance through each classifier head and choosing one or more classifier head predictions based on e.g. entropy, may be useful. Here we solve this problem in a supervised manner.

A common workflow in industry is to train a deep ANN on a large number of samples from some modality and domain, then transfer the copious information in its learned representations to downstream tasks. Transferring to a downstream task can leave the deep ANN's parameters unchanged and instead involve training a single new layer (or "head") atop the hidden states of the deep ANN. In this scenario, the deep ANN is pre-trained on task $t_0$ and transfer learning means training on task $t_1$. Optimal performance on downstream tasks sometimes requires updating the parameters of the deep ANN, however, which prevents parameter reuse across downstream tasks and raises the inference costs of large-scale deployments. EideticNets address this problem by enabling a new workflow: train the deep ANN and, for every downstream task $t_i : i \in \{1, \ldots, K\}$ that requires updating the deep ANN, train the deep ANN as an EideticNet to ensure that (1) all of its learned representations are preserved on the original pre-training task, (2) the downstream tasks are as accurate as possible, and (3) parameter reuse is preserved and operational expenses are minimized.

The parallel between transfer learning and continual learning that we draw in the previous paragraph highlights an important distinction between EideticNets and some other approaches to catastrophic forgetting. Other approaches (Kirkpatrick et al., 2017; Ritter et al., 2018; Zenke et al., 2017) see the scope of catastrophic forgetting as encompassing both representation space and classification space. EideticNets are designed with representation space in mind and ensure that forgetting is impossible in representation space. Thus, where some approaches to mitigating catastrophic forgetting have a single classifier head for all tasks in their reported evaluations, EideticNets have one head per task.

---

[2]The sparsity pattern in Figure 2b is the same as that introduced in Golkar et al. (2019) and was discovered independently by the authors of this manuscript.

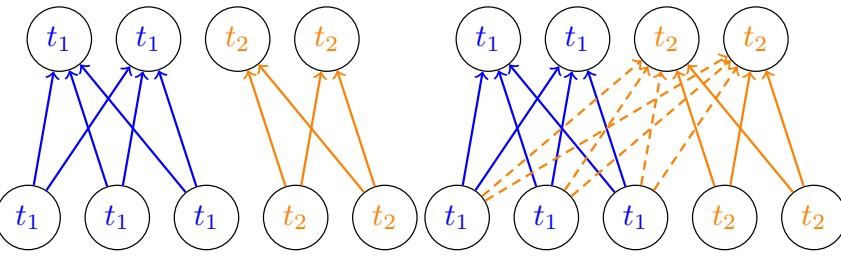

(a) Without nesting, features are disjoint across tasks.

(b) Nesting efficiently reuses an ANN's capacity.

Figure 2: Eidetic Learning eliminates forgetting by preserving important neurons, deleting unimportant synapses, and recycling unimportant neurons for subsequent tasks. Preserving important neurons can be done in several ways. Figure a depicts a network in which the neurons of task $t_2$ are completely separated from the neurons of task $t_1$. This is an inefficient use of a network's capacity. Since task $t_2$ is trained after task $t_1$, allowing the neurons important to $t_2$ to benefit from the features learned by $t_1$ in the previous layer is more efficient. This latter way is shown as the dashed orange lines in Figure b. A detailed depiction along with the parameters' configurations is also provided in figs. 5a and 5b.

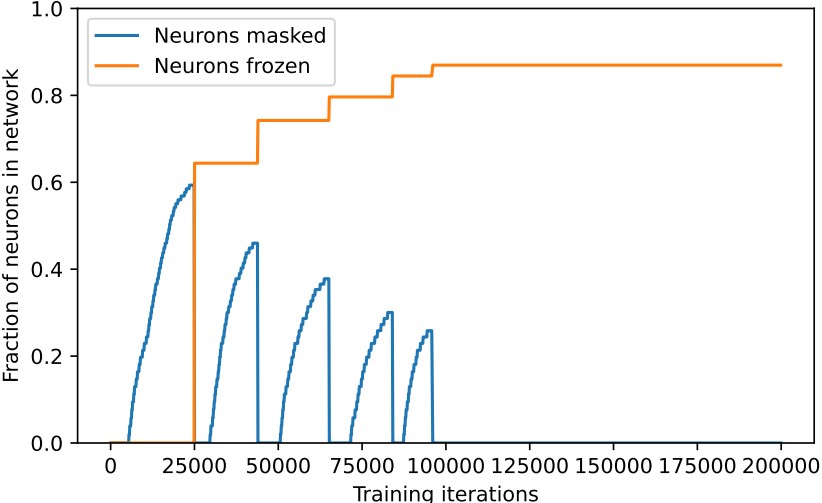

Figure 3: In Eidetic Training, the fraction of neurons pruned while training a task increases until training set accuracy drops, and the unpruned neurons are frozen cumulatively. Figure shows the progress of training with a ResNet50 trained on Sequential CIFAR100 with five tasks.

Ours is a practical approach with strong guarantees that supports contemporary ANN architectures without modification, whereas Kaushik et al. (2021) requires a separate batch normalization layer per task.

Since EideticNets have only a few hyperparameters, and do not require changes to the loss function, they are in practice quite easy to train. We have developed a PyTorch framework that makes creating an EideticNet a straightforward task: when writing the class that defines your network, simply subclass our framework's `EideticNetwork` class and specify the relationships among your network's layers. An example of code that trains an EideticNet on multiple tasks is shown in Listing 1. Calling `train_task` (line 6) automatically selects the minimal subset of neurons necessary to train a task without loss of accuracy. This code works out of the box without changes to the network architecture, optimizer, or loss function.

```
1    model = MyEideticNet()
2
3    for t, train_loader in enumerate(train_loaders):
4        model.prepare_for_task(t)
5        optimizer = optim.AdamW(model.parameters(), lr)
6        model.train_task(
7            train_loader, optimizer=optimizer, ...
8        )
```

Listing 1: Example of training an EideticNet using our framework.

There are numerous other benefits to EideticNets.

- Efficiency: The time and memory complexity of EideticNets is linear in the number of parameters and training a task required no rehearsal or replay.

- Robustness: An EideticNet can be trained on significantly different subsequent tasks without harming performance on existing tasks and – unlike regularization approaches to catastrophic forgetting – without requiring a additional hyperparameter search just to preserve the tasks for which the network has already been trained.

- Stability: A neuron is either important or not for the tasks trained so far and its importance persists across tasks, resulting in a stable training procedure.

- Interpretability: For each new task, EideticNets account for the incremental amount of capacity required, so calculating the amount of excess capacity remaining for any given layer in the network is straightforward.

- Maintainability: a layer in an EideticNet can be widened to add more capacity when needed, because there are no synaptic connections from unimportant neurons in layer $\ell$ to important neurons in layer $\ell + 1$.

## 2 NOTATIONS AND RELATED WORK

**Notations.** Define a neural network as a function $f_\theta \in \mathcal{F} : \mathcal{X} \to \mathbb{Z}$ from function class $\mathcal{F}$ with training data $(\mathbf{x}_i^0, y_i) \in (\mathcal{X}, \mathbb{Z})$ parameterized by $\theta := \{\mathbf{W}^\ell : \ell \in \{1, \ldots, L\}\}$ with each $\mathbf{W}^\ell \in \mathbb{R}^{N_\ell \times D_\ell}$, where $N_\ell$ is the number of neurons of layer $\ell$ and $D_\ell$ the input dimensions. Note that $D_\ell = N_{\ell-1}$. Let $\sigma$ be a pointwise function. Without loss of generality, let $\sigma(\mathbf{x}) = \mathbf{x}$. Consider the setting in which a neural network is trained on tasks $t_i : i \in \{1, \ldots, K\}$. We denote the parameters after training task $t_j$ as $\theta_{t_j}$. Let $\mathcal{N}$ denote the set of all neurons in an ANN and $\mathcal{N}_{t_i}$ the smallest set of neurons necessary to perform task $t_i$.

For a neural network $f_{\theta_{t_j}}(\mathbf{x}_i)$ to be immune to forgetting, it must guarantee the following conditions for all tasks $t_i$ and layers $\ell$.

(I) **Persistence**: The neurons in layer $\ell$ important to task $t_i$ should remain unchanged during the training of any subsequent task $t_{j>i}$.

(II) **Resistance**: The neurons in layer $\ell - 1$ that are not important to $t_i$ no longer affect the $t_i$-important neurons in layer $\ell$ once training of $t_i$ is complete.

For any task $t_i$, the persistence condition preserves the weights necessary to perform the task and the resistance condition ensures that the hidden states entering any layer remain the same $\forall \mathbf{x}_i \in \mathcal{X}$. Note that the resistance condition is transitive. These conditions are sufficient to guarantee the immutability of $f_{\theta_{t_j}}(\mathbf{x}_i)$ during training of any subsequent task. Catastrophic forgetting is thus solved by any method that guarantees these conditions.

**Related works.** Catastrophic forgetting is a much-studied problem. Previous work has exploited the sparsity pattern shown in Figure 2b. Golkar et al. (2019) introduce Continual Learning via Neural Pruning (CLNP). Using this sparsity pattern, they report state of the art results on Permuted MNIST with a feed-forward network. We show in Table 3 that our method is competitive. The complexity of the number hyperparameters of CLNP is linear in the number of layers of a network. Similarly,

Mallya et al. (2018) start with a pre-trained network and learn one separate binary mask, using a hyperparameter to control the degree of sparsity, for each layer and task. Mallya & Lazebnik (2018) induce sparsity via iterative pruning with structured weight magnitude pruning. We contrast our method with this previous work in Table 1. Our method only requires a constant number of hyperparameters per ANN, which makes it feasible for real-world use cases, we investigate pruning with scoring on a network-wide normalized scale (Molchanov et al., 2017), instead of per layer, and provide insights into the interplay between continual learning and normalized scoring. Further, our method does not require selecting a task ID before performing the forward pass through the ANN. Finally, the implementation of our method is comprehensive and open source, and is tightly integrated with PyTorch's pruning.

| Method | Open source | Hyperparameters | Pruning | Forward transfer | Task IDs |
|---|---|---|---|---|---|
| Piggyback | ✓ | Per-layer and task | Weight | ✗ | Required |
| PackNet | ✓ | Per-network | Per-task masks | ✓ | Required |
| CNLP | ✗ | Per-layer | $\ell_1$ penalty | ✓ | Required |
| Ours | ✓ | Per-network | Weight, Taylor | ✓ | Not required |

Table 1: Comparison of our method to previous sparsity-inducing work on catastrophic forgetting. No methods support backward transfer of features to previous tasks. We describe an extension to Eidetic Learning in Section 5 to support backward transfer.

Jung et al. (2020) proposes a regularization-based strategy where consecutive tasks are associated to important and unimportant neurons. After a task is learned and its important neurons identified, a regularizer helps to maintain the connection to the important neurons of previous task to prevent forgetting. It's unclear whether this works with standard optimization algorithms, as the work employs proximal gradient descent. Overall, this approach aims for the same goal as Eidetic Training, but the guarantees of our method are stronger.

Kirkpatrick et al. (2017) intermittently estimate the importance of parameters via the Fisher information of the gradients and, during subsequent training, regularize the parameters that are important to previous tasks to remain close to their pre-trained values. Other methods compute the importance of parameters online rather than intermittently (Ritter et al., 2018; Zenke et al., 2017). Kao et al. (2021) investigates catastrophic forgetting in recurrent networks and proposes to combine regularization and projected gradient descent.

A approach that similarly enables subsets of neurons to specialize on specific tasks is the sparsely-gated mixture of experts layer (MoE) Shazeer et al. (2016). An MoE layer is a set of discrete layers, each one trained to specialize in a task or subset of tasks in the training data. The MoE layer is equipped with a data-conditional routing module that determines which expert has specialized in a particular input $x$. An MoE layer and an EideticNet resemble one another in that they support data-conditional routing. An EideticNet has the advantage of enabling feature reuse across tasks.

Other approaches have been proposed to exploit sparsity in useful ways. Most notable among them, in relation to our work, are NestedNet (Kim et al., 2018) and Russian Doll Networks (Jiang & Mu, 2021). NestedNet allows for the identification of nested subnetworks within an ANN such that the innermost network is the smallest and fastest, thereby enabling the selection, for a fixed task $t_1$, of the subnetwork with the latency and memory characteristics most suited to a target deployment environment. For both of these approaches, the nesting is within layers. With EideticNets, the nesting occurs *across* layers and subsequent tasks $t_{j>i}$. Evci et al. (2022) describe Head2Toe, an approach to multitask learning that selects from a pre-trained network the neurons that are important for a given task.

## 3 EIDETIC LEARNING: A PRINCIPLED SOLUTION TO CATASTROPHIC FORGETTING

We now sketch the procedure for training an EideticNet on tasks $\{t_1, \ldots, t_K\}$ and identify the steps that enable EideticNets to satisfy both the persistence and resistance conditions. Starting with an empty set $\mathcal{F} = \emptyset$ for recording important neurons at each step, task $t_{i+1}$ is trained with no impact on previously-trained task $t_i$ thus:

I Via iterative pruning, find $\mathcal{N}_{t_i}$, the smallest set of neurons required to perform task $t_i$ without loss of accuracy.

II **Persistence**: Update the set of frozen neurons $\mathcal{F} = \mathcal{F} \cup \mathcal{N}_{t_i}$.

III Let $\mathcal{R} = \mathcal{N} \setminus \mathcal{F}$ be the set of neurons to recycle when training $t_{i+1}$. (These neurons were pruned in step I of this procedure.)

IV Reinitialize the neurons in $\mathcal{R}$.

V **Resistance**: Prune the synaptic connections $(r, n_{t_i}) : \{r \in \mathcal{R} \wedge n \in \mathcal{N}_{t_i}\}$ only if $r$ is a neuron in $\mathbf{W}^\ell$ and $n_{t_i}$ is a neuron in $\mathbf{W}^{\ell+(k \geq 1)}$.

VI Train $t_{i+1}$ in a conventional manner.

Freezing the important neurons for $t_i$ in step II ensures that EideticNets satisfy the persistence condition. Before proceeding to train task $t_{i+1}$, we reinitialize the unimportant neurons that were pruned in step I. To satisfy the resistance condition, we prune their synaptic connections to neurons in $\mathcal{F}$ in downstream layers. Once pruned, those synaptic connections are guaranteed to remain so, because the connections are pruned by setting to 0 an input dimension of a neuron in the frozen set $\mathcal{F}$. Both conditions are satisfied and EideticNets thus cannot forget.

### 3.1 Layers of an Eidetic Network

We now describe how EideticNets are implemented within the layers of an ANN. At the end of this section, we also propose an argument for why making self-attention layers of Transformers (Vaswani et al., 2017) immune to catastrophic forgetting is challenging if not impossible.

**Linear**    The linear layer simply applies the following operation to its input feature map $\boldsymbol{x}$

$$Linear(\boldsymbol{x}) = \boldsymbol{W}\boldsymbol{x} + \boldsymbol{b}, \tag{1}$$

where the bias $\boldsymbol{b}$ is optional, e.g., removed when that layer is followed by batch-normalization. During training, both parameters $\boldsymbol{W}, \boldsymbol{b}$ are trained. To extract excess capacity, we employ structured pruning on $\boldsymbol{W}$ by zeroing-out multiple rows based on our pruning strategy. However, this is not enough to ensure that subsequent training tasks will not impact the current task. In fact, we also need to ensure that the remaining nonzero entries of $\boldsymbol{W}$ do not interact with already pruned units from the previous layer–since those will be re-initialized and trained in subsequent tasks. As such, we also need to propagate the previous layer's pruning to the columns of $\boldsymbol{W}$. Pruning of the bias vector $\boldsymbol{b}$ is done simply by using the output unit pruning mask of $\boldsymbol{W}$.

**Convolution**    The convolutional layer is also a Linear layer, as in Equation (1), but with a special structure on the parameters $\boldsymbol{W}$ and (optionally) $\boldsymbol{b}$. In particular, if the input to the convolutional layer is flattened as a vector $\boldsymbol{x}$ then the convolution operation makes $\boldsymbol{W}$ a circulant-block-circulant matrix. In that setting, the structured pruning and previous layer pruning propagation is applied to the channel dimensions. That is, during pruning, we select which of the output channels filters to entirely prune, i.e., all the *input_channels* $\times$ *width* $\times$ *height* parameters for each *output_channel* filter. Then, we also need to pruning the *input_channels* filters, i.e., for all the *output_channels* $\times$ *width* $\times$ *height* parameters based on the previous layer pruning.

**Batch normalization**    Recall that batch normalization (BN) Ioffe & Szegedy (2015) shifts and scales the entries of the its input using four additional parameters $\boldsymbol{\mu}, \boldsymbol{\sigma}, \boldsymbol{\beta}, \boldsymbol{\gamma}$. Define $x_k$ as $k^{\text{th}}$ entry of feature map $\boldsymbol{x}$ and $\mu_k, \sigma_k, \beta_k, \gamma_k$ as the $k^{\text{th}}$ entries of the BN parameter vectors $\boldsymbol{\mu}, \boldsymbol{\sigma}, \boldsymbol{\beta}, \boldsymbol{\gamma}$, respectively. Then we can write the BN layer mapping as

$$BN(\boldsymbol{z})_k = \frac{\boldsymbol{z}_k - \mu_k}{\sigma_k} \gamma_k + \beta_k. \tag{2}$$

The parameters $\boldsymbol{\mu}, \boldsymbol{\sigma}$ are computed as the element-wise mean and standard deviation of $\boldsymbol{z}_\ell$ during each mini-batch learning step, and as their moving average at evaluation time. The parameters $\boldsymbol{\beta}, \boldsymbol{\gamma}$ are learned.

In EideticNets, a BN layer is handled precisely as follows. The learnable parameters $(\boldsymbol{\beta}, \boldsymbol{\gamma})$ are pruned and frozen according to the previous linear or convolutional pruning mask. Normally, during

training of subsequent tasks, the statistics $\boldsymbol{\mu}, \boldsymbol{\sigma}$ will continue to adapt to each new task. To guarantee perfect preservation of the previously-trained tasks in BN layers, we also to (i) make the $\boldsymbol{\beta}, \boldsymbol{\gamma}$ of the previous tasks stay in evaluation mode when training subsequent tasks, and (ii) ensure that $\boldsymbol{\beta}, \boldsymbol{\gamma}$'s internal running statistics are not updated.

This guarantees that BN layers perfectly retain statistics specific to a task $t_i$ and do not drift during subsequent training. This is one of the key benefits of our method and our choice to employ structured pruning makes this possible.

**Residual connections**    It is common to add residual connections to deep ANNs to aid optimization. Our strategy works the same regardless of the actual architecture of the internal block. Consider the following case:

$$\boldsymbol{R}\boldsymbol{z} + \mathrm{ReLU}(\mathrm{BN}^{(2)}(\boldsymbol{W}^{(2)}\mathrm{ReLU}(\mathrm{BN}^{(1)}(\boldsymbol{W}^{(1)}\boldsymbol{z})))). \tag{3}$$

Hence, a typical residual block with two nonlinear layers. The internal block pruning for $\boldsymbol{W}^{(2)}, \boldsymbol{W}^{(1)}, \mathrm{BN}^{(2)}, \mathrm{BN}^{(1)}$ should follow the previously introduced rule. But then, we also need to handle $\boldsymbol{R}$ carefully as otherwise the output of the residual layer would shift with subsequent task training (that would change parts of the output distribution of $\boldsymbol{z}$). To ensure that we preserve our guarantees, we can simply prune $\boldsymbol{R}$ as follows. Use the input unit mask of $\boldsymbol{W}^{(1)}$ for the input units of $\boldsymbol{R}$ (which are made so that the task separation is respected by construction), and use the output unit mask of $\boldsymbol{W}^{(2)}$ (or $\mathrm{BN}^{(2)}$ since they are the same) for the output units of $\boldsymbol{R}$. That construction holds regardless of the use of nonlinearity in Equation (3), the use of BN, or the number of internal layers–as long as the first and last layers' constraints are the ones ported to $\boldsymbol{R}$.

**Recurrent and LSTM layers.**    Our proposed solution can be implemented with recurrent and gated models out-of-the-box. In fact, all those models involve Linear operations (recall Equation (1)) and element-wise nonlinearites and gates. Hence, once simply need to ensure that the relationship between adjacent linear layers is known, e.g., in a vanilla RNN this would be between the input-hidden matrix and the hidden-hidden recurrent matrix within a layer, and between the hidden-hidden recurrent matrix of a layer,and the input-hidden matrix of the next layer, and ensure that the constrained are respected between them. then, the use of the nonlinearity will not impact the results. An evaluation of recurrent networks is out of scope for the current work?

## 4    Experiments

In all experiments in this paper, we employed structured pruning. On each pruning iteration, we selected the same percentage of neurons in each layer to prune. The selected neurons are those with the lowest score. The scoring function is fixed during a particular experiment. Neuron scores are either the $p$-norm, $p \in \{1, 2\}$, or the Taylor score (Molchanov et al., 2017). To limit the time required for iterative pruning, we limited the maximum number of training epochs to recover from one pass of pruning to a fixed number, and we designed the recovery loop to exit when training set accuracy is within the threshold. All models are trained with the nested features of Figure 2b.

To select hyperparameters, we hold out 10% of the training set, train for three tasks, and choose the set of hyperparameters with the best held-out set accuracy averaged across the three tasks. The crucial hyperparameters are the pruning step size and the (training set accuracy) threshold for when to stop pruning. The models and datasets with which we perform experiments, and the pruning hyperparameters we searched for each, are shown in Table 2.

All networks were trained using the Adam optimizer and no weight decay on a single NVIDIA V100 GPU with 16 GB of on-chip memory.

Model architectures are as defined in Section 6.1.

**Permuted MNIST**    To evaluate our approach on the Permuted MNIST (PMNIST) dataset, we use the same setup as (Golkar et al., 2019) and others: a 2-hidden layer feed-forward network with 2000 hidden units, a learning rate of 0.002, the Adam optimizer, and a batch size of 256.

| Model | Dataset | # Tasks | Step sizes | Stop thresholds | Batch size | Learning rate |
|-------|---------|---------|-----------|-----------------|-----------|---------------|
| MLP | Permuted MNIST | 10 | 1%, 5% | .1%, 1% | 256 | 2e-4 |
| ResNet18 | Imagenette | 5 | 1%, 5% | .3%, 3% | 32 | 1.25e-5 |
| ResNet50 | CIFAR-100 | 5 | 1%, 5% | .3%, 3% | 256 | 1e-4 |

Table 2: Experimental setup and pruning hyperparameters used during hyperparameter search. The step size is the percentage of neurons to prune in one pruning iteration and the stop threshold is the percentage below the maximum-achieved training set accuracy at which iterative pruning stops. Once the stop threshold is reached, the weights from the previous pruning iteration are restored, and Eidetic Learning moves onto the next task.

The hyperparameter grid for this task included Dropout (Srivastava et al., 2014) with probabilities $\{0, .03, .10\}$, due to the width of the layers. The grid also included whether to reduce the learning rate by half between training and pruning. The best hyperparameters found when evaluating by holding out 10% of the training set were: Taylor pruning, a step size of 5%, a stop-pruning threshold of 0.1%, and a dropout probability of 0.1, and reducing the learning rate by half before starting to prune. The results of the hyperparameter search for PMNIST are shown in Table in the appendix.

When training the final model, we used early stopping with a patience of 10 epochs on the training set accuracy and a maximum number of 5 recovery epochs. The results of the final model are shown in Figure 1. Once a task has been trained, accuracy on it remains constant as subsequent tasks are trained. The competitiveness of our method on PMNIST is shown in Table 3.

Table 3: Mean and standard deviation of our method on 10 tasks of Permuted MNIST with 2000 neurons in each of the 2 hidden layers.

| Method | Accuracy (%) |
|--------|--------------|
| Single Task SGD | $98.48 \pm 0.05$ |
| Kirkpatrick et al. (Kirkpatrick et al., 2017) | 97.0 |
| Zenke et al. (Zenke et al., 2017) | 97.2 |
| Cheung et al. (Cheung et al., 2019) | 97.6 |
| Golkar et al. (Golkar et al., 2019) | $98.42 \pm 0.04$ |
| Ours | $98.31 \pm 0.09$ |

**Deep networks**    We trained a deep residual network, ResNet50, with Sequential CIFAR100 with 10 tasks. We were only able to evaluate $\ell_1$ and $\ell_2$ weight magnitude pruning in this setting, due to the tendency of Taylor pruning not to prune the layers of a network uniformly. We discuss this future in the final section of this manuscript. The results are shown in Table 4.

**High-resolution images**    To show the scalability of our method to high-resolution images, we evaluate our method on $224 \times 224$ images from Imagenette (Howard, 2019), a subset of images from the ImageNet dataset (Deng et al., 2009). The images belong to 10 classes that are easy to classify accurately. As such, they are likely to be classified correctly early in training. We evaluate on the dataset in a sequential setting, with classes grouped in pairs across 5 tasks.

We trained ResNet50 with a batch size 32 with learning rate 1.25e-5. We fixed the early stopping patience for pre-training at 10 epochs and did not reduce the learning rate after pre-training. The maximum number of recovery training epochs during iterative fine-tuning was 10. We show per-task accuracy in Table 5 averaged across three runs with different random seeds. We observe that

Table 4: Mean and standard deviation over 3 independent runs of accuracy of ResNet50 on Sequential CIFAR100 with 5 tasks.

| Task 0 | Task 1 | Task 2 | Task 3 | Task 4 |
|--------|--------|--------|--------|--------|
| 80% (±0.01) | 78% (±0.02) | 75% (±0.03) | 70% (±0.04) | 75% (±0.03) |

Table 5: Mean and standard deviation over 3 independent runs of accuracy of ResNet18 on Sequential Imagenette with 5 tasks.

| Task 0 | Task 1 | Task 2 | Task 3 | Task 4 |
|--------|--------|--------|--------|--------|
| 88% (±0.00) | 67% (±0.10) | 81% (±0.07) | 84% (±0.01) | 77% (±0.08) |

EideticNets are able to produce strong per-class performances, even when learning the last task, i.e., when the remaining excess capacity is reduced from the previous 4 tasks.

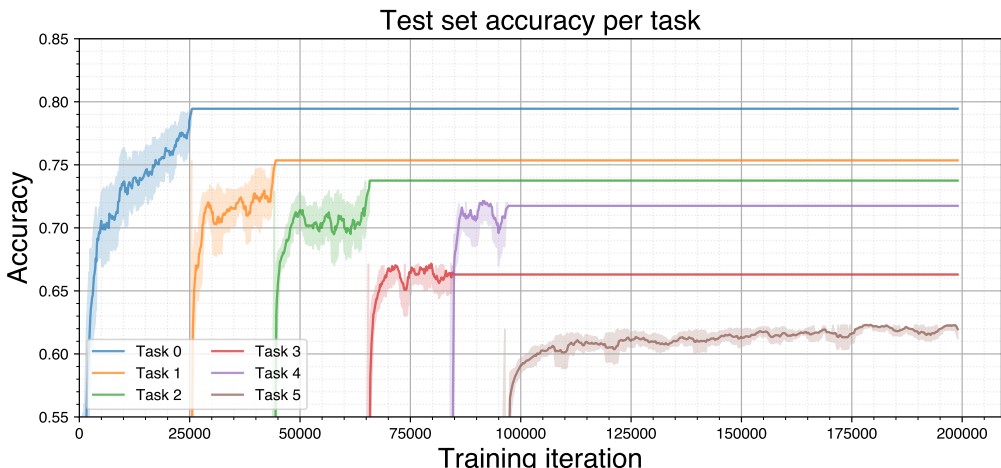

Figure 4: Test set accuracy of ResNet50 trained on Sequential CIFAR100 with five tasks.

**Effect of task classifier on per-class accuracy** As mentioned, EideticNets do not require knowledge of the task ID at inference time. This is in sharp contrast with other methods and we believe this is more aligned with real-world scenarios. EideticNets effectively assign the task ID of a new instance. To illustrate the impact of this capability on performance, we compare task routing with an EideticNet to oracle task routing using a small and a large network. Table 7 shows MNIST-5 and CIFAR10-5 and with a small MLP trained with Eidetic learning using $\ell_1$, $\ell_2$, and Taylor pruning, the evolution of the per-class performances with the trained task router, against an oracle task router. As shown, the per-class performance varies if the task router had perfect test performance. Table 8 shows task routing results with ResNet50 an CIFAR100-5 in which the average drop in per-class accuracy is 1.8%. There is indeed a performance gap to be closed by better task routing mechanisms, especially when going to more complex tasks (CIFAR10-5). While Eidetic Learning already provides competitive performance in their current realization, we believe that this is an interesting avenue for future research.

## 5 CONCLUSIONS AND FUTURE WORK

Aspects of our method are incompatible with some desiderata of a continual learning method.

*Forward transfer* of features allows subsequent tasks $t_{j>i}$ to benefit from features learned during training of a previous task $t_i$ (cf. Figure 2b). *Backward transfer* allows previous tasks to benefit from subsequently-learned features. Eidetic Learning, as presented, only supports forward transfer. It can be straightforwardly extended to support backward transfer as follows: to enable backward transfer from $t_2$ to $t_1$, apply Eidetic Learning after training $t_2$ to free excess neurons, then re-train $t_1$ and only allow updates to the newly-freed neurons and to the $t_1$ classifier head. This preserves the previously-trained neurons of $t_1$ and allows the new neurons of $t_1$ to benefit from the neurons of $t_2$.

Whether this requires replay using the data on which $t_1$ was initially trained depends on whether the data distribution is stationary.

In *task-incremental learning*, tasks are learned in a discrete sequence of phases and classes do not overlap between tasks. In *class-incremental learning* (CIL), classes can overlap between tasks (Masana et al., 2022; Gummadi et al., 2022). Eidetic Learning currently assumes a *task incremental learning* setting and does not address *class-incremental learning*. Future work may extending Eidetic Learning to support class-incremental learning and other more challenging settings in the future.

Some of the experimental setups we report results on in this paper could only be run with weight magnitude pruning. We observed that Taylor pruning (Molchanov et al., 2017) does not uniformly prune across layers. It tends to prune neurons towards a network's output first. Consequently, it's possible for a task to be trained and pruned to the point of its training set accuracy falling below the stop threshold before some capacity in each layer has been pruned. This violates the principles we enumerated in Section 3. On the other hand, weight magnitude pruning of a fixed percentage of neurons across all layers of a network is indiscriminate and, we believe, results in some excess capacity in some layers of a network not being pruned. Future work on hybrid approaches that use Taylor and weight magnitude pruning may result in more compact and more accurate networks.

We have presented Eidetic Learning and demonstrated – with a variety of architectures and combinations of datasets – that they exploit an ANN's excess capacity to prevent catastrophic forgetting. Once an EideticNet learns a task, it retains it perfectly. We advise the reader of inherent constraints on the effective use of EideticNets. Training one to good effect requires that excess capacity exists in the ANN. Excess capacity is a function of several factors, such as a neural network's ambient dimensionality and number of layers as well as the complexity of the task implied by the dataset. A given neural network may have excess capacity with respect to one task $t_i$ but not to another $t_j$. We advise prudent adoption of EideticNets and hope they are of benefit in industry, academic, and other contexts.

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

# 6 SUPPLEMENTARY MATERIAL

Authors may wish to optionally include extra information (complete proofs, additional experiments and plots) in the appendix. All such materials should be part of the supplemental material (submitted separately) and should NOT be included in the main submission.

## 6.1 MODELS USED IN THIS STUDY

- **MLP (Permuted MNIST only)**: 2 hidden layers with 2000 neurons in the input and hidden layers.
- **ResNet18**: ResNet18 with one additional convolutional layer for each skip connection.
- **ResNet50**: ResNet50 adapted for low-resolution images for evaluation on Sequential CIFAR100. The first layer has filters of size $3 \times 3$ instead of $7 \times 7$, and the first max pooling layer is removed.

To support pruning, the ResNets used in this study have one additional convolutional layer per skip connection. A standard ResNet has skip connections of the form $f(x) + x$. These networks have $f(x) + g(x)$ and the extra convolutional layer $g$ is pruned to jointly match the sparsity of the output of a block $f(x)$ and the layer that produced $x$.

Table 6: Hyperparameter sweep results of ResNet18 with Imagenette

| Average accuracy | Pruning type | Pruning step size | Stop threshold | Early stopping patience | Frozen |
|---|---|---|---|---|---|
| 74.61 | 2 | 0.01 | 0.00 | 10 | 0.95 |
| **83.96** | 2 | 0.01 | 0.03 | 10 | 0.27 |
| 75.18 | 2 | 0.05 | 0.00 | 10 | 0.90 |
| 83.21 | 2 | 0.05 | 0.03 | 10 | 0.32 |

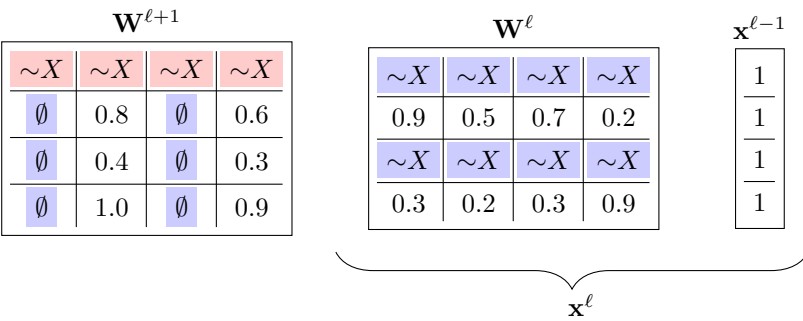

(a) Eidetic Network without feature sharing. A synapse (as an edge in the undirected the network graph) is deleted if it connects a neuron that is unimportant for task $t_i$ to a neuron that is important for task $t_i$.

(b) Eidetic Network with feature sharing. A synapse (as an edge in the directed of the network graph) is deleted if it connects a neuron that is unimportant for task $t_i$ to a neuron that is important for task $t_i$.

Figure 5: Consider a feed-forward ANN with layers $\ell$, $\ell + 1$ trained on some task $t_i$. Omitting for convenience the non-linearity $\sigma$ and the bias $\boldsymbol{b}$, processing the input $\boldsymbol{x}^{\ell-1}$ vector of all 1s entails the matrix-vector products $\boldsymbol{W}^\ell \boldsymbol{x}^{\ell-1}$ and $\boldsymbol{W}^{\ell+1} \boldsymbol{x}^\ell$. We show them here as the composition $\boldsymbol{W}^{\ell+1}(\boldsymbol{W}^\ell \boldsymbol{x}^{\ell-1})$. Imagine that the smallest set of neurons required to perform $t$ is determined to be $\mathcal{N}_t := \{\boldsymbol{W}_2^\ell, \boldsymbol{W}_4^\ell, \boldsymbol{W}_2^{\ell+1}, \boldsymbol{W}_3^{\ell+1}, \boldsymbol{W}_4^{\ell+1}\}$ (white). For task $t_i$, the excess capacity consists of all other neurons, and the neurons to recycle, $\mathcal{R}$ when training $t_{i+1}$ are $\{\boldsymbol{W}_1^\ell, \boldsymbol{W}_3^\ell, \boldsymbol{W}_1^{\ell+1}\}$ (blue in $\boldsymbol{W}^\ell$, red in $\boldsymbol{W}^{\ell+1}$). While training $t_i$, we prune the neurons $\mathcal{R}$ and permanently delete their synaptic connections to the important neurons (blue $\emptyset$s in $\boldsymbol{W}^\ell$). When training of $t_i$ is complete, we reinitialize the neurons in $\mathcal{R}$ from some random variable $X$. Figure a illustrates the naive approach that leads to the complete partitioning of task $t_i$ from $t_{j>i}$ (cf. Figure 2a). The efficient nested feature sharing that EideticNets enable is shown in Figure b (cf. Figure 2b).

Table 7: Per-class accuracy with learned task classifier (top row) and oracle task classifier (bottom row) for different dataset and pruning methods, using a small MLP model. For each group, the top row is the EideticNet performance.

| | | | | | | | | | | | |
|---|---|---|---|---|---|---|---|---|---|---|---|
| $\ell_1$ | mnist | 99.39 | 99.74 | 97.77 | 97.92 | 99.08 | 98.32 | 98.12 | 98.15 | 97.74 | 96.83 |
| | | 99.69 | 99.74 | 98.06 | 98.32 | 100 | 98.32 | 98.75 | 98.44 | 97.84 | 97.82 |
| | cifar10 | 63.40 | 64.70 | 36.30 | 34.30 | 54.30 | 44.30 | 54.10 | 47.80 | 62.00 | 63.40 |
| | | 80.60 | 71.50 | 76.60 | 38.90 | 79.20 | 53.40 | 94.20 | 54.50 | 79.90 | 66.60 |
| $\ell_2$ | mnist | 99.29 | 99.47 | 97.77 | 99.41 | 98.17 | 97.53 | 98.85 | 99.22 | 95.38 | 97.22 |
| | | 99.80 | 99.74 | 97.87 | 99.70 | 99.29 | 97.87 | 99.37 | 99.42 | 95.69 | 98.81 |
| | cifar10 | 55.10 | 68.90 | 43.40 | 35.00 | 36.00 | 38.70 | 67.00 | 60.70 | 65.60 | 65.10 |
| | | 79.70 | 72.40 | 80.40 | 38.80 | 70.90 | 56.50 | 93.90 | 66.20 | 85.40 | 66.60 |
| taylor | mnist | 98.88 | 99.56 | 99.22 | 98.02 | 98.57 | 97.87 | 98.23 | 98.54 | 97.43 | 97.22 |
| | | 99.29 | 99.56 | 99.42 | 98.61 | 99.29 | 97.98 | 99.37 | 99.22 | 98.97 | 98.12 |
| | cifar10 | 54.90 | 68.00 | 36.60 | 30.30 | 44.60 | 41.60 | 58.70 | 60.40 | 53.80 | 64.50 |
| | | 69.20 | 89.60 | 61.60 | 64.30 | 64.70 | 61.70 | 77.80 | 73.50 | 80.00 | 64.50 |

Table 8: Per-class accuracy of ResNet50 with CIFAR-10 with oracle task (Oracle) routing and with task routing via a learned task classifier (Eidetic). The average reduction in accuracy using task routing is -1.8%. The average accuracy of oracle and task routing are 56.4% and 54.6%, respectively.

| Class | Oracle | Eidetic | Delta | Class | Oracle | Eidetic | Delta |
|---|---|---|---|---|---|---|---|
| 00 | 76.33 | 76.00 | -0.33% | 50 | 44.67 | 43.67 | -1.0% |
| 01 | 70.33 | 69.67 | -0.67% | 51 | 53.00 | 52.00 | -1.0% |
| 02 | 41.67 | 40.67 | -1.0% | 52 | 63.67 | 63.33 | -0.33% |
| 03 | 32.67 | 32.67 | 0.0% | 53 | 80.33 | 79.33 | -1.0% |
| 04 | 44.33 | 42.00 | -2.3% | 54 | 69.67 | 69.33 | -0.33% |
| 05 | 60.00 | 54.67 | -5.3% | 55 | 34.33 | 28.33 | -6.0% |
| 06 | 64.67 | 62.67 | -2.0% | 56 | 71.67 | 71.33 | -0.33% |
| 07 | 61.67 | 55.00 | -6.7% | 57 | 63.00 | 62.67 | -0.33% |
| 08 | 69.33 | 68.00 | -1.3% | 58 | 70.33 | 70.00 | -0.33% |
| 09 | 72.67 | 70.33 | -2.3% | 59 | 53.33 | 51.00 | -2.3% |
| 10 | 39.33 | 33.67 | -5.7% | 60 | 77.33 | 76.33 | -1.0% |
| 11 | 44.33 | 40.33 | -4.0% | 61 | 56.00 | 53.00 | -3.0% |
| 12 | 71.67 | 65.33 | -6.3% | 62 | 54.67 | 54.33 | -0.33% |
| 13 | 52.00 | 51.00 | -1.0% | 63 | 51.67 | 49.67 | -2.0% |
| 14 | 46.67 | 45.67 | -1.0% | 64 | 34.00 | 31.67 | -2.3% |
| 15 | 64.00 | 63.00 | -1.0% | 65 | 29.67 | 27.67 | -2.0% |
| 16 | 64.67 | 63.67 | -1.0% | 66 | 59.00 | 57.67 | -1.3% |
| 17 | 73.00 | 72.67 | -0.33% | 67 | 44.00 | 38.67 | -5.3% |
| 18 | 55.00 | 54.00 | -1.0% | 68 | 80.33 | 79.67 | -0.67% |
| 19 | 53.00 | 52.00 | -1.0% | 69 | 68.00 | 67.67 | -0.33% |
| 20 | 77.67 | 75.67 | -2.0% | 70 | 60.67 | 60.33 | -0.33% |
| 21 | 76.33 | 72.33 | -4.0% | 71 | 66.67 | 64.33 | -2.3% |
| 22 | 51.00 | 49.33 | -1.7% | 72 | 27.00 | 24.33 | -2.7% |
| 23 | 74.33 | 74.00 | -0.33% | 73 | 37.00 | 35.67 | -1.3% |
| 24 | 66.00 | 64.00 | -2.0% | 74 | 42.00 | 37.67 | -4.3% |
| 25 | 44.33 | 41.67 | -2.7% | 75 | 66.33 | 66.00 | -0.33% |
| 26 | 45.33 | 44.67 | -0.67% | 76 | 77.33 | 76.67 | -0.67% |
| 27 | 42.00 | 39.33 | -2.7% | 77 | 44.00 | 43.00 | -1.0% |
| 28 | 70.67 | 70.33 | -0.33% | 78 | 51.33 | 45.33 | -6.0% |
| 29 | 54.33 | 54.00 | -0.33% | 79 | 57.00 | 55.00 | -2.0% |
| 30 | 57.67 | 55.00 | -2.7% | 80 | 31.33 | 28.00 | -3.3% |
| 31 | 53.67 | 52.67 | -1.0% | 81 | 64.33 | 64.33 | 0.0% |
| 32 | 53.67 | 52.67 | -1.0% | 82 | 76.33 | 76.33 | 0.0% |
| 33 | 48.00 | 43.00 | -5.0% | 83 | 49.00 | 48.33 | -0.67% |
| 34 | 63.67 | 61.00 | -2.7% | 84 | 37.33 | 34.33 | -3.0% |
| 35 | 41.00 | 40.33 | -0.67% | 85 | 59.00 | 58.00 | -1.0% |
| 36 | 65.67 | 63.67 | -2.0% | 86 | 56.33 | 55.33 | -1.0% |
| 37 | 54.67 | 54.33 | -0.33% | 87 | 59.67 | 59.33 | -0.33% |
| 38 | 43.33 | 41.67 | -1.7% | 88 | 51.67 | 51.33 | -0.33% |
| 39 | 73.00 | 71.00 | -2.0% | 89 | 58.67 | 58.00 | -0.67% |
| 40 | 54.33 | 51.67 | -2.7% | 90 | 53.67 | 53.33 | -0.33% |
| 41 | 71.00 | 70.67 | -0.33% | 91 | 60.67 | 58.00 | -2.7% |
| 42 | 64.00 | 63.67 | -0.33% | 92 | 49.67 | 46.33 | -3.3% |
| 43 | 68.33 | 66.33 | -2.0% | 93 | 29.33 | 26.67 | -2.7% |
| 44 | 28.33 | 26.67 | -1.7% | 94 | 85.00 | 84.33 | -0.67% |
| 45 | 37.00 | 34.00 | -3.0% | 95 | 54.67 | 54.00 | -0.67% |
| 46 | 42.33 | 40.33 | -2.0% | 96 | 47.33 | 46.00 | -1.3% |
| 47 | 48.33 | 47.00 | -1.3% | 97 | 55.33 | 54.67 | -0.67% |
| 48 | 86.33 | 85.00 | -1.3% | 98 | 34.00 | 31.33 | -2.7% |
| 49 | 69.67 | 68.00 | -1.7% | 99 | 56.33 | 51.67 | -4.7% |

