# OpenReview forum: "Eidetic Learning: an Efficient and Provable Solution to Catastrophic Forgetting"
_ICLR.cc/2025/Conference — ICLR 2025 Conference Withdrawn Submission_

### Official Review · Reviewer_uoMk · 2024-10-18

**Soundness:** 3
**Presentation:** 3
**Contribution:** 3
**Rating:** 6
**Confidence:** 1

**Summary:**

This paper introduces a framework called EideticNet, a novel approach aimed at solving the problem of catastrophic forgetting in neural networks. The approach utilizes a network’s excess capacity to ensure that once a task is learned, it is not forgotten, regardless of subsequent tasks. The methodology involves iterative pruning and selective freezing of neurons critical to previously learned tasks, thus preserving their functionality while continuing to train on new tasks.

**Strengths:**

1. The introduction of Eidetic Learning provides a fresh perspective on addressing catastrophic forgetting by leveraging the concept of memory retention through structural network adjustments.

**Weaknesses:**

**I am not familiar with related fields. After reading the paper, I can probably understand the motivation and technical content.**

---

1. There is a lack of ablation study, recent baselines (post-2020) and the well-time cost of EideticNet required.

2. The authors do not conduct large-scale experiments on ImageNet 1K or transformer-based networks.

**Questions:**

See weaknesses.

---

### Official Review · Reviewer_VhRR · 2024-10-30

**Soundness:** 1
**Presentation:** 3
**Contribution:** 2
**Rating:** 3
**Confidence:** 4

**Summary:**

This paper introduces Eidetic Learning, a novel continual learning method aimed at eliminating catastrophic forgetting. Eidetic Learning, implemented in neural networks called EideticNets, operates by iteratively pruning less important neurons for a given task. The weights of the remaining, important neurons are then frozen, ensuring that subsequent training on new tasks does not alter their functionality. A central claim made by the paper is that it “provably solves catastrophic forgetting”. This claim seems to be derived from the fact that, when moving to the next task, the method avoids training connections that could affect the activity of neurons considered important for the performance on previous tasks. Indeed, the pruned neurons are reinitialized and their connections towards the frozen neurons are severed, allowing this freed capacity to be used for learning the next task without interfering with previously learned representations. Experiments on Permuted MNIST with fully connected networks show that Eidetic Learning achieves accuracy competitive with state-of-the-art methods like Golkar et al. The method's scalability is also assessed on Imagenette and CIFAR-100 with ResNet architectures, although no comparative results are provided for these datasets. To facilitate adoption, the authors plan to release a PyTorch implementation of their EideticNet framework.

**Strengths:**

Originality: The paper demonstrates a degree of originality by explicitly highlighting the importance of severing connections from reinitialized (previously pruned) neurons to frozen neurons crucial for prior tasks. While pruning has been explored in continual learning before, this precise mechanism for guaranteeing resistance to interference and preserving performance on past tasks appears novel.

Quality: The paper presents a reasonably clear method with an intuitive justification for its ability to prevent forgetting. The experimental validation on the Permuted MNIST benchmark, while limited in scope, goes some way towards supporting the central claim on addressing catastrophic forgetting. The reporting of confidence intervals in the results are appreciated. The commitment to releasing a PyTorch library also enhances the potential for reproducibility.

Clarity: The paper is reasonably well-written and explains the Eidetic Learning procedure in a concise manner.

Significance: The paper addresses an important problem, that of the significant challenge of catastrophic forgetting in continual learning, a problem with substantial implications for developing more adaptable and robust AI systems. The proposed method, with its guarantees and relatively simple implementation, has the potential to influence future research in this area. If the empirical results generalize to a wider range of tasks and architectures, Eidetic Learning could become a valuable ingredient for practitioners seeking to deploy continual learning models in real-world applications. The promised open-source library could contribute to adoption as well.

**Weaknesses:**

Unfortunately, I have a number of concerns regarding the readiness for publication of this work, touching on all aspects of originality, quality, clarity and significance.

Originality: Generally, the originality of the work is quite limited. The idea of leveraging sparsity of network connectivity for continual learning is certainly not unheard of. Notably, prior work “Compacting, Picking and Growing for Unforgetting Continual Learning” (NeurIPS 2019) seem to propose the same mechanism of pruning weights learned so far to free capacity for training the pruned weights on future tasks. However, that work doesn’t seem to showcase the “resistance” mechanism that ensures there is no backward interference with previous tasks, as in this submission.

Quality: The paper makes a number of fairly bold claims on the importance of the work, but doesn’t provide a lot of experimental comparisons with prior work, providing comparisons on only 1 dataset, which I would argue falls short of the experimentation quality standard of ICLR.

Clarity:
* The paper mentions that “We also propose an argument for why making self-attention layers of Transformers (Vaswani et al., 2017) immune to catastrophic forgetting is challenging if not impossible”. However, I don’t think the paper actually does that… is it missing from the text?
* The paper mentions that the method “trains a final task classifier on a meta task dataset”. But, doesn’t this require actually knowing the total number of tasks that are coming, if the task classifier is a softmax (i.e. multi-class) classifier? Generally, more clarity regarding how the corresponding experiments were conducted would be appreciated.
* The paper claims that “an EideticNet can be trained on significantly different subsequent tasks without harming performance on existing tasks and – unlike regularization approaches to catastrophic forgetting – **without requiring a additional hyperparameter search** just to preserve the tasks for which the network has already been trained”. However, don’t you need to specify how much pruning you’re willing to do, i.e. how low the accuracy can drop on previous tasks? Wouldn’t this be a hyper-parameter to tune?
* The paper also has a number of typos and unclear statements:
  * seteting => setting
  * a additional hyperparameter => an additional hyperparameter
  * A approach => An approach
  * we also need to pruning => we also need to prune
  * we also to => we also need to
  * An evaluation of recurrent networks is out of scope for the current work? => change “?” with “.”
  * “r is a neuron in Wl and nti is a neuron in Wl+(k≥1)” => but W is not a set of neuron, but a matrix of weights… so “r is a neuron in W” doesn’t really make sense
  * delete the synapses from pruned to unpruned neurons (directionally) => not sure what this is supposed to mean….

Significance:
 * A central claim made in the paper is that “Eidetic Learning [...] provably solves catastrophic forgetting”. However, taken literally, this statement is inaccurate, and at a minimum exaggerated. First, prior work such as on “Progressive Neural Networks” by Rusu et al. (2016) has already “solved” catastrophic forgetting, simply by adding neurons for each new task and keeping previous neurons fixed. The authors might argue that their work is an improvement because it does not require the explicit addition of neurons, however this implies that, on a large scale with lots of tasks, EideticNets could eventually saturate and have no capacity left for new tasks, making them an incomplete solution to the problem of continual learning.
 * The limited experiments presented does not allow me to confirm that this work is a significant departure from the state-of-the-art, since it presents a comparison on only one dataset and the performance is statistically indistinguishable from the prior work of Golkar et al.

**Questions:**

I would appreciate it if the authors could answer the questions mentioned in the weakness section, notably regarding clarity.

I would also like the authors to report additional experimental comparisons based on at least one other dataset. For example, the authors could do a comparison on the CIFAR-100 20 tasks setting found in “Compacting, Picking and Growing for Unforgetting Continual Learning”.

The paper would also require rewording to soften some of its claims, though unfortunately this would require submitting a revision, which I don’t think the ICLR review process allows for.

---

### Official Review · Reviewer_ajYR · 2024-10-31

**Soundness:** 2
**Presentation:** 1
**Contribution:** 1
**Rating:** 1
**Confidence:** 4

**Summary:**

The paper proposes a method for continual learning that prunes weights after each task is trained. The pruned weights are `disconnected’ from the active nodes for a task, such that later training of the pruned weights doesn’t influence previous tasks. This allows to have zero forgetting, but loses capacity for later tasks. To prevent the requirement of a task-id at inference, a task classifier is trained which is used to select the appropriate head at inference time. The proposed method is tested on several benchmarks.

**Strengths:**

* Pruning is an approach in continual learning that makes sense, so it is good to further research it.
* The experiments that are carried out make sense
* What is in the paper is well written, although some important information is missing

**Weaknesses:**

* The proposed method relies on task-id prediction to select the appropriate head. Except for benchmarks where there is a clear difference between the tasks, it is not possible to train a network that can predict a task-id, without already being able to predict the class itself. Let’s assume two simple tasks with e.g. horses and cats in task one, and deer and dog in task two. If a task-id predictor would be able to tell that an example belongs to task one, it must also know that it is either a horse or a cat. The inverse would imply that you know that the you don’t know whether it is a horse or a cat, but do know that it is definitely not a deer or a dog. Only when the first two have some common characteristic this is possible, e.g. when the first task would all be vehicles with wheels and the second one all animals on four legs. For instance, in permuted MNIST this is true, as the permutation mask is shared within a task, but for the other benchmarks there is no stronger relation between classes within a task than across tasks hence the task-id prediction is not feasible. To convince me that task-id prediction is possible it would be good to at least show the accuracy of the task predictor and compare it with how well the individual classes are separated in the representation (e.g. by linear probing the representation with the entire dataset).

* Several key elements in the paper are almost not explained or discussed, while other more technical details are discussed at length. It would make the paper better to move section 3.1 to the appendix, and instead discuss details of the pruning method and how the task-id prediction works.  The technical details are relevant if someone would want to reimplement the method, but they don’t learn me a lot of how and why the method works. Right now, there is barely any information about how those work, while they are the most important aspects of the proposed method. Similarly, the ablations that are discussed in the paragraph of line 463 are important results to learn about the method and thus it would be better to condense those tables and add them to the main paper.

* There is only little comparison to other methods that rely on similar techniques. This is problematic, as the proposed method is very similar to e.g. PackNet. Table 1 lists some differences, but it is unclear how they make the proposed algorithms actually different. For instance, PackNet masks out activations of neurons that belong to later tasks, while here the weights between those connections are removed. That is a difference, but only a technical one that doesn’t impact the results. Only for Permuted MNIST there is a comparison with some other methods, while all the other benchmarks are not compared to other methods. Other methods that are very similar are not considered (PackNet and Piggyback). It would have been insightful to test those methods both with and without the proposed task-id prediction mechanism. In the current paper, there is no proof that the proposed method works any better than older methods.

* The paragraph at line 349 contains main spelling and grammar mistakes, as well as a question. It seems like this is from a draft version. At the start of section 3 (line 291), there is a sentence about attention layers, but the referred text is not included as far as I can tell.

**Questions:**

* Do you have any justification for why the task-id predictor should work?
* Why did you not compare the results to similar methods, like PackNet?

---

### Official Review · Reviewer_Au6C · 2024-11-03

**Soundness:** 3
**Presentation:** 2
**Contribution:** 2
**Rating:** 3
**Confidence:** 4

**Summary:**

The authors propose the Eidetic Learning method aimed at solving catastrophic forgetting. This approach leverages the concept of EideticNet, a neural network architecture that allocates and reuses neurons through structured pruning. By freezing neurons critical to earlier tasks and reinitializing less important ones for subsequent tasks, it effectively ensures task retention without the need for rehearsal or replay.

**Strengths:**

The use of structured pruning in Eidetic Learning for addressing catastrophic forgetting is unique. This theoretically sound solution offers clear guarantees for preventing forgetting. Additionally, the authors plan to release a PyTorch framework, which will facilitate adoption and further experimentation by the research community.

**Weaknesses:**

While the proposed method shows competitive performance against the compared baselines, it only evaluates against other approaches on PMNIST and does not include comparisons with CIFAR-100 or Imagenette. This lack of a comprehensive evaluation makes it hard to measure the true efficacy of Eidetic Learning.

Furthermore, there is no comparison using larger datasets like ImageNet-100 or Tiny ImageNet, which again makes it impossible to assess how the method performs with datasets containing more than 10 classes, the maximum number tested in the study.

**Questions:**

Did the authors attempt to convert EideticNet to use transformers or attention mechanisms? I ask because most state-of-the-art (SOTA) methods are based on attention models.

Similarly to Table 3, could the authors provide comparable results for CIFAR-100 and Imagenette in Tables 4 and 5, respectively?

The authors claim that EideticNet is efficient and robust, but without experimentation on large-scale datasets, this claim is unsubstantiated. How would EideticNet perform against other approaches on datasets with 100 classes like ImageNet-100 or 200 classes like Tiny ImageNet?

---

### Note · Authors · 2024-12-04

**Comment:**

We thank the reviewers for their time and helpful feedback and apologize for not having responded during the rebuttal period. We acknowledge the limitations of the current evaluation and results. We nonetheless believe the method we describe in our paper has untapped potential and will continue to develop it.

**Withdrawal Confirmation:**

I have read and agree with the venue's withdrawal policy on behalf of myself and my co-authors.